# Code-Generated Graph Representations Using Multiple LLM Agents for Material Properties Prediction

Jiao Huang [1 2]   Qianli Xing [1 3]   Jinglong Ji [1 2]   Bo Yang [1 3]

## Abstract

Graph neural networks have recently demonstrated remarkable performance in predicting material properties. Crystalline material data is manually encoded into graph representations. Existing methods incorporate different attributes into constructing representations to satisfy the constraints arising from symmetries of material structure. However, existing methods for obtaining graph representations are specific to certain constraints, which are ineffective when facing new constraints. In this work, we propose a code generation framework with multiple large language model agents to obtain representations named Rep-CodeGen with three iterative stages simulating an evolutionary algorithm. To the best of our knowledge, Rep-CodeGen is the first framework for automatically generating code to obtain representations that can be used when facing new constraints. Furthermore, a type of representation from generated codes by our framework satisfies six constraints, with codes satisfying three constraints as bases. Extensive experiments on two real-world material datasets show that a property prediction method based on such a graph representation achieves state-of-the-art performance in material property prediction tasks.

## 1. Introduction

Over the past decades, machine learning methods have been widely used in material science. These methods normally employ hand-crafted descriptors as the representation of

---

[1]Key Laboratory of Symbolic Computation and Knowledge Engineering of Ministry of Education, Jilin University, Changchun, Jilin, 130012, China [2]College of Artificial Intelligence, Jilin University, Changchun, Jilin, 130012, China [3]College of Computer Science and Technology, Jilin University, Changchun, Jilin, 130012, China. Correspondence to: Qianli Xing <qianlixing@jlu.edu.cn>, Bo Yang <ybo@jlu.edu.cn>.

*Proceedings of the $42^{nd}$ International Conference on Machine Learning*, Vancouver, Canada. PMLR 267, 2025. Copyright 2025 by the author(s).

material data (Damewood et al., 2023). Recently, graph neural networks (GNN) have shown remarkable performance in predicting material properties and become the dominant technology (Chen et al., 2024b; Fang & Yan, 2024).

For most GNN-based material properties predicting methods, a crystalline material data instance is represented by an attributed graph, where nodes represent the atoms and the edges define the interatomic reactions of neighboring atoms (Xie & Grossman, 2018). The attributes of the graph commonly contain elemental properties as node features and interatomic distances as edge features. With the manually designed codes, the attributed graph can be translated into a numerical form, called a graph representation, which can be processed by GNN. As many important material properties are highly sensitive to the structure of materials, a fundamental challenge for these methods is to obtain the representation that can capture the structure of crystalline material(Choudhary & DeCost, 2021).

A widely adopted solution by researchers is to incorporate different attributes into the process of constructing the graph and the representation. CGCNN (Xie & Grossman, 2018), Matformer (Yan et al., 2022), and PotNet(Lin et al., 2023) mainly focus on the distance of atoms, while ALIGNN(Choudhary & DeCost, 2021), PerCNet(Huang et al., 2025), and ComFormer(Yan et al., 2024) further incorperates angle attributes. With the different attributes, representations satisfy different constraints arising from symmetries of material structure. Ideally, representations capture structures better when satisfying more constraints. However, existing methods for obtaining graph representations are specific to certain constraints, which is ineffective when facing new constraints. Moreover, the above research strategy requires expert knowledge in both material and computer science, which costs researchers years to learn.

Inspired by the achievements of large language models (LLM) in code generation and material science, we propose a code generation framework with multiple LLM agents to obtain representations named Rep-CodeGen. The codes generated by large language models are always of low quality when the task is difficult. Specifically, the main difficulty is to guide LLM agents to generate codes to obtain representation satisfying constraints, which requires LLM agents

to accomplish the complex design of the material graph construction process with material knowledge and the coding process with computer science knowledge. Thus, we simulate the process of an evolutionary algorithm by dividing the overall framework into three stages: crossover generation, evaluation summary, and parent selection. In crossover generation stages, we incorporate two agents: the first agent is used to take the prompts and generate plans on how to write the codes, and the second agent is used to generate codes according to plans. Then, in the evaluation summary stage, we score the generated codes with a test dataset and use the third agent to provide a detailed analysis. Finally, in the parent selection stage, we set a metric to select the codes with high scores as parent codes for the next loop. The main contributions of this paper are as follows:

- We propose an interpretable framework for automatically generating codes to obtain graph representations that can be used when facing new constraints.

- We obtain a type of graph representation from generated codes by our framework satisfying six constraints with codes satisfying three constraints as bases.

- Extensive experiments in two real-world material datasets show that a property prediction method based on such graph representation achieves state-of-the-art performance in material property prediction tasks.

## 2. Related Works

### 2.1. Material Representation with Deep Learning

In recent decades, significant advancements in machine learning have been achieved in the study of crystalline materials (Meredig et al., 2014; Oliynyk et al., 2016; Ward et al., 2016; Ramprasad et al., 2017). For material property prediction task, the development of representations that satisfy constraints of crystalline material structures is crucial for success. CGCNN (Xie & Grossman, 2018) introduced a multigraph-based approach, leveraging periodicity and invariance under rotation and translation by encoding atomic interactions through multiple edges between nodes. Subsequent studies have expanded on this foundation by incorporating additional physical insights. For example, SchNet (Schütt et al., 2017) integrates force data into training, MegNet (Chen et al., 2019) includes thermodynamic variables like temperature and pressure, and ALIGNN (Choudhary & DeCost, 2021) utilizes dual graph structures to represent bond lengths and angles. To better capture the contributions of individual atoms within a unit cell, GATGNN (Louis et al., 2020) employs attention mechanisms to model local atomic interactions. MatFormer (Yan et al., 2022) advances this by introducing a framework that ensures periodic invariance, while PotNet (Lin et al., 2023) models interatomic

potentials through infinite distance summations.

Subsequently, to ensure that the graph representations of material structures satisfy reflectional symmetry, methods such as ComFormer(Yan et al., 2024) and PercNet(Huang et al., 2025) extended the multigraph representation by introducing higher-dimensional features, including bond angles and dihedral angles. These enhancements further improved the prediction accuracy of machine learning algorithms in material property prediction tasks.

To satisfy Lipschitz continuity, AMD(Widdowson et al., 2022) and PDD(Widdowson & Kurlin, 2022) proposed modeling materials as periodic point clouds. Unlike CGCNN (Xie & Grossman, 2018), which represents bond lengths as fixed scalars, these methods suggest using distance distributions to model bond lengths.

With the advancement of large language models (LLMs), leveraging their generative capabilities across various domains has emerged as a new research direction. Works such as material LLM (Tao et al., 2024) and MatExpert (Ding et al., 2024) propose modeling material generation tasks directly as string generation tasks, producing materials represented in CIF format. However, these works cannot theoretically guarantee that the generated material structures satisfy physical symmetry constraints, and the results lack interpretability. This significantly undermines the trust of domain experts in LLMs and even neural network outputs, limiting the practical application of LLMs in the field of materials. In contrast, we propose a novel approach to applying LLMs in the material domain. Specifically, instead of directly generating representations, our method generates interpretable code. Human experts can understand the underlying principles of the results through the code, thereby enhancing trust in the outputs of LLMs.

The statistics of constraint fulfillment for the aforementioned algorithms are provided in Table 3 in the Appendix.

### 2.2. Multi-agent System

To better leverage the potential of large language models (LLMs), multi-agent systems often use specific frameworks to connect and invoke multiple LLMs, enabling the completion of tasks that are more complex or specialized than what a single LLM can achieve.

Due to the confabulations (or hallucinations) (Romera-Paredes et al., 2024) inherent in LLM outputs, incorporating evaluators to verify LLM-generated results is a common approach in agent systems. For instance, Haluptzok et al. (Haluptzok et al., 2022) employs a Python interpreter to filter the correctness of data generated by the language model, thereby enhancing its performance. Similarly, Zelikman et al. (Zelikman et al., 2022) adopts this idea by validating previously generated data (such as explanations for answers)

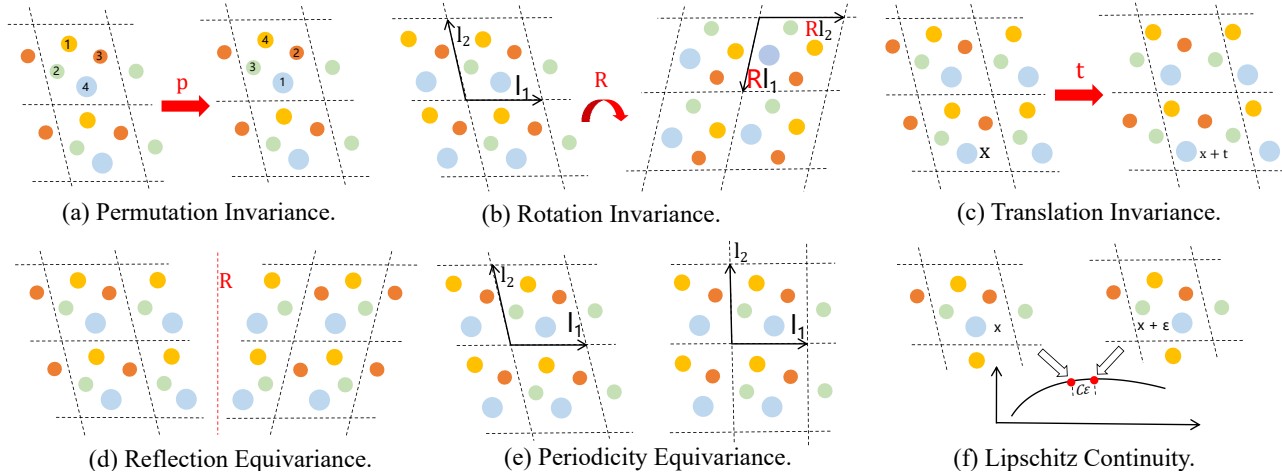

(a) Permutation Invariance.  (b) Rotation Invariance.  (c) Translation Invariance.

(d) Reflection Equivariance.  (e) Periodicity Equivariance.  (f) Lipschitz Continuity.

*Figure 1.* Constraints of Crystalline Material Structures. For simplicity, we use a 2D structure for illustration. (a) Permutation Invariance: The atomic indices within the unit cell change. (b) Rotation Invariance: The lattice and all atomic coordinates undergo the same rotation. (c) Translation Invariance: All atomic coordinates undergo the same translation. (d) Reflection Equivariance: The material structure is mirrored along a specific plane. (e) Periodicity Equivariance: The lattice coordinates change. (f) Lipschitz Continuity: The atomic coordinate $x$ undergoes a small displacement $\epsilon$.

to ensure that the LLM can be fine-tuned on correctly generated answers. Romera-Paredes et al. (Romera-Paredes et al., 2024) utilizes a code executor to test LLM-generated code, ensuring the accuracy of the generated data.

More relevant to our approach is the use of LLMs as a source of mutations in genetic programming processes. Haluptzok et al. (Haluptzok et al., 2022) was the first to propose the concept of "evolution through large models", leveraging code-generating LLMs as mutation operators in genetic programming (GP). This idea was subsequently applied to various fields, including neural architecture search (Chen et al., 2024a; Zheng et al., 2023; Nasir et al., 2024), code generation (Romera-Paredes et al., 2024; Lehman et al., 2023), symbolic regression (Meyerson et al., 2024), and game exploration (Wang et al., 2023). Unlike previous tasks, applying LLMs in real-world domains requires careful consideration of physical rules and the generation of interpretable results to enhance trustworthiness. To the best of our knowledge, we are the first to apply the concept to the field of material representation, proposing a representation of material structures that surpasses current human-designed representations.

## 3. Preliminaries

### 3.1. Material Notations

A material's structure can be viewed as the infinite extension of its unit cell, with the unit cell serving as the smallest representative entity illustrating the material's structure. A material can be expressed as $M = \{A, X, L\}$.

Here, $A = [a_1, ..., a_N]^\top \in \mathbb{A}^N$ denotes atom types, with $\mathbb{A}$ representing the set of chemical elements and $N$ representing the number of atoms in the unit cell. $X = [x_1, ..., x_N]^\top \in \mathbb{R}^{N \times 3}$ specifies the three-dimensional coordinates of atoms in the Cartesian coordinate system. $L = [l_1, l_2, l_3]^\top \in \mathbb{R}^{3 \times 3}$ represents the periodic lattice, indicating the directions in which the unit cell extends infinitely in three-dimensional space.

### 3.2. Constraints of Crystalline Material Structures

We consider six widely recognized constraints that material molecular representations should satisfy: permutation invariance, rotation invariance, reflection equivariance, lipschitz continuity, periodicity equivariance, and translation invariance. The illustrations of these constraints are shown in Figure 1. Their detailed definitions are provided below.

**Permutation Invariance:** For any permutation $p \in S_N$, we have $P(A, X, L) = P(Ap, X, L)$. This means that changing the order of the atoms does not affect the representation of the material structure.

**Rotation Invariance:** For any rotation matrix $R \in \mathbb{R}^{3 \times 3}$, we have $P(A, X, L) = P(A, XR, LR)$. This indicates that applying the same rotation to both the atoms and the lattice does not alter the representation of the material structure.

**Translation Invariance:** For any translation vector $t \in \mathbb{R}^{3 \times 1}$, we have $P(A, X, L) = P(A, X + t1^T, L + t1^T)$, where $1 \in \mathbb{R}^{1 \times 3}$. This means that translating both the atoms and the lattice by the same vector does not change the representation of the material structure.

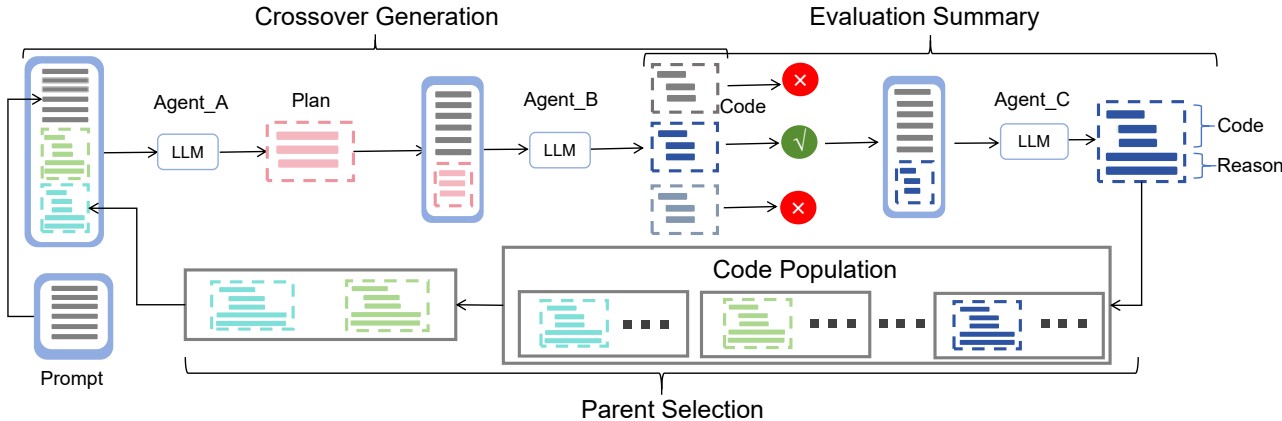

*Figure 2.* Pipeline of code generation framework with multiple LLM agents to obtain representations (Rep-CodeGen).

**Reflection Equivariance:** For any reflection matrix $R \in \mathbb{R}^{3 \times 3}$, we have $P(A, X, L) \neq P(A, XR, LR)$. This implies that performing the same reflection on both the atoms and the lattice will change the representation of the material structure.

**Periodicity Equivariance:** For any orthogonal matrix $O \in \mathbb{R}^{3 \times 3}$, if $L_2 \neq OL_1$, then $P(A, X, L_1) \neq P(A, X, L_2)$. This indicates that changing the lattice coordinates of the material structure will change the representation.

**Lipschitz continuity:** If the material structure $m_2$ is obtained by shifting each atom's position in $m_1$ by at most $\epsilon$, then $d(P(m_2), P(m_1)) \leq C\epsilon$, where $C$ is a constant. This means that the change in the representation of the material structure should be continuous.

## 4. Methodology

In this section, we first introduce a code generation framework with multiple LLM agents to obtain graph representations named Rep-CodeGen. An overview of Rep-CodeGen is shown in Figure 2. We simulate the process of an evolutionary algorithm by dividing the overall framework into three stages: Crossover Generation, Evaluation Summary, and Parent Selection. We will describe each component in Sections 4.1, 4.2, and 4.3, respectively.

### 4.1. First Stage: Crossover Generation

The purpose of the first phase, the Crossover Generation, is to generate new material structure representation codes based on existing parent codes using Large Language Models (LLMs). To further improve the quality and interpretability of the generated output, we adopted a two-step generation strategy. In the first step, the plan for solving the problem is generated by $Agent\_A$. In the second step, a complete code is generated based on the outlined plan by $Agent\_B$. Below, we describe the detailed process of each step.

#### 4.1.1. SOLUTION GENERATION

The goal of solution generation is to utilize the $Agent\_A$ to devise a plan for generating material representation code that satisfies all constraints, based on the physical constraints of the parent code. In this section, the input to the $Agent\_A$ mainly consists of three parts: the problem description, the parent code, and the task requirements. An example template of such a prompt is shown in the Appendix D.1.

In the first part of the prompt, the problem description, we begin by providing the $Agent\_A$ with a natural language explanation of the task, followed by a brief description of the constraints to be satisfied. Then, we present all the environmental code in the form of code, with the core being an evaluate function that scores the generated representation code. The second part, the description of the parent code, includes the code of the parent code, the satisfaction of the constraints, and the potential reasons for each constraint being either satisfied or unsatisfied. These reasons and the satisfaction status are obtained from the code evaluation and $Agent\_C$ analysis in the second stage, evaluation summary. Finally, we describe the task to $Agent\_A$.

The $Agent\_A$ serves as the creative core of our evolutionary framework, generating plans for new mutation code by referencing the parent code and their constraint satisfaction status, along with the reasons provided in the prompt. Based on this information, $Agent\_A$ attempts to generate a plan for code that satisfies all constraints.

### 4.1.2. CODE GENERATION

The goal of code generation is to use $Agent\_B$ to generate material representation code that satisfies all constraints, based on the modification plan generated by $Agent\_A$. Similar to the previous section on solution generation, the input to the $Agent\_B$ still consists of three parts: the problem description, the parent code, and the task requirements. An example template of such a prompt is shown in D.2.

Here, the problem description and the description of the parent code are the same as in the previous prompt for $Agent\_A$. The third part, the task description for the $Agent\_B$, incorporates the output from $Agent\_A$, i.e., the modification plan, as part of the input. Each iteration generates four codes, after which all the generated code is evaluated and analyzed.

### 4.2. Second Stage: Evaluation Summary

The purpose of the second stage, Evaluation Summary, is to assess the constraint satisfaction of the generated code using the test data, and to utilize $Agent\_C$ to analyze the reasons behind the satisfaction or violation of these constraints. The detailed process of each step is described below.

The code is evaluated and scored using a constructed dataset. During the evaluation process, the input to the evaluation function consists of test data for each constraint, and the output is the score of the code for each constraint. We have built dedicated test data for each constraint based on the MP-20 dataset, with the construction process and the scoring criteria for each constraint detailed below. The construction of test sets for each constraint is shown in Appendix C. To accelerate the evaluation process, a random subset of data is sampled from the constructed test set for each evaluation.

If the code passes all test cases for a specific constraint, we consider it to satisfy that constraint; otherwise, it is deemed unsatisfied. Finally, incorrect codes (those that fail to execute within the specified time and memory limits, or produce invalid outputs) will be discarded. The correctly running code, along with its constraint satisfaction status, will be sent to the $Agent\_C$ for cause analysis.

To enable $Agent\_A$ to have a more accurate and deeper understanding of the parent code, and to generate a modification plan that better meets the requirements, we use the LLM to analyze the reasons for the code satisfying or violating certain constraints before sending the generated code to the code repository. Similar to the previous prompt for the agent, the input to $Agent\_C$ still consists of three parts: the problem description, the code with its constraint satisfaction status, and the task requirements. An example template of such a prompt is shown in D.3.

Finally, the satisfaction status of the code on the various constraint are used as labels, and the code, along with its reasons, are stored in the code population.

### 4.3. Third Stage: Parent Selection

The main purpose of code population is twofold. First, it stores the codes obtained from the previous step of the Evaluation Summary along with the reasons for their corresponding constraint satisfaction. Second, it serves as a source for parent codes for the first stage, Crossover Generation.

Parent selection, which refers to the process of sampling codes from the code population to generate prompts in Crossover Generation, is a crucial component of this framework. For this, we cluster codes based on their signatures, with codes having the same label being placed in the same cluster. The signature of the $i$-th code is defined as $S^i = [s_1^i, \ldots, s_K^i]$, where $K$ is the number of constrains, and $s_k^i \in \{0, 1\}$ represents the satisfaction of the $k$-th constraint by the $i$-th code within its respective cluster. Specifically, $s_k^i = 0$ indicates that the code's output does not satisfy the $k$-th constraint, while $s_k^i = 1$ indicates that the constraint is satisfied. The number $n$ represents the total number of constraints, which is set to 6 by default.

To ensure diversity in sampling and comprehensive constraint satisfaction, we prioritize selecting two codes that satisfy a larger number of constraints. For this purpose, we propose a joint label for two clusters, $S^{i,j} = [s_1^{i,j}, \ldots, s_K^{i,j}]$, where $K$ is the number of constrains, and the definition of $s_k^{i,j}$ is as follows: if $s_k^i = 1$ or $s_k^j = 1$, then $s_k^{i,j} = 1$; otherwise, $s_k^{i,j} = 0$.

Based on this, the probability that the $i$-th cluster and the $j$-th cluster are jointly selected as parent codes is given by:

$$P_{i,j} = \frac{\exp\left(\frac{\overline{S^{i,j}}}{T}\right)}{\sum_{i'=1}^{N} \sum_{j'=1}^{N} \exp\left(\frac{\overline{S^{i',j'}}}{T}\right)},$$

$$\overline{S^{i,j}} = \frac{1}{K}\sum_{k=1}^{K} s_k^{i,j}, \quad \overline{S^{i',j'}} = \frac{1}{K}\sum_{k=1}^{K} s_k^{i',j'}, \quad (1)$$

where, $N$ is the total number of clusters, $K$ is the number of constrains, and $T$ is the temperature parameter related to the current number of codes. Following Boltzmann distribution (Maza & Tidor, 1993), a code pair with a larger $S^{i',j'}$ (i.e., satisfies more constraints) is more likely to be selected.

Considering the token limitations of the input and output of the large model, when selecting parent codes from a cluster, we tend to prefer shorter codes. In this case, the probability of selecting the $i$-th code within a cluster is given by:

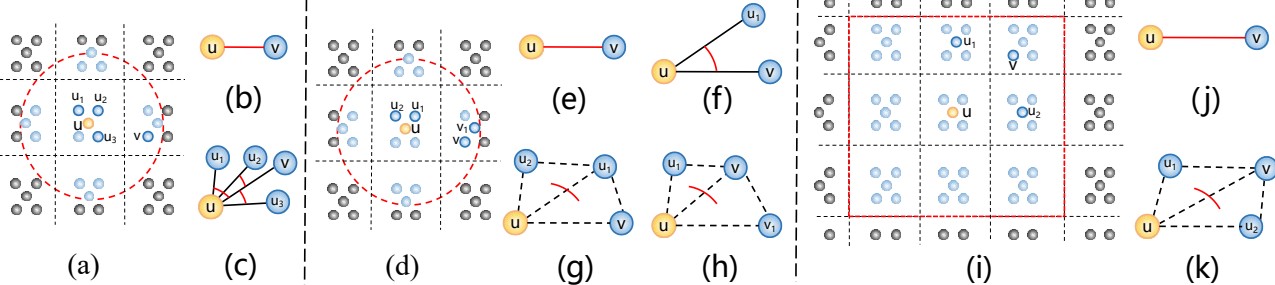

*Figure 3.* Illustrations of graph representation constructed by ComFormer (a-c), PerCNet (d-h) and ours (i-k). For simplicity, we use a 2D structure for illustration. (a) Illustration of ComFormer's cutoff-based neighbor selection, where the red dashed circle (with a radius equal to the cutoff value) encloses all atoms (blue nodes) that are neighbors of atom $u$ (orange node), and black nodes represent non-neighbor atoms. (b–c) Illustrations of ComFormer's distance and angle features, where $u_1$, $u_2$, and $u_3$ are the three nearest neighbors of atom $u$ and $v$ is the neighbor of atom $u$. (d) Illustration of PerCNet's cutoff-based neighbor selection. (e–h) Illustrations of PerCNet's distance, angle, and dihedral angle features, where $u_1$ is the nearest neighbor of atom $u$, $u_2$ is the neighbor with the smallest angle relative to the edge $uv$, and $v_1$ is the nearest neighbor of atom $v$. (i) Illustration of our proposed lattice-based neighbor selection, where all atoms (blue nodes) within the red dashed box are neighbors of atom $u$ (orange node). (j–k) Illustrations of our proposed distance and dihedral angle features, where $u_1$ and $u_2$ are atoms obtained by extending atom $u$ along the lattice directions.

$$P_i = \frac{\exp\left(-\frac{C_i}{T}\right)}{\sum_{i'=1}^{N_c} \exp\left(-\frac{C_{i'}}{T}\right)}, \qquad (2)$$

where, $N_c$ is the total number of codes in the cluster, and $C_i$ is the character count of the $i$-th code. With Equation 1 and 2, we achieve the goal that code pairs satisfying more constraints and having shorter lengths are more likely to be selected as parent codes.

## 5. Experiments

In this section, we address three key questions: (**Q1**) Can our framework identify a representation that satisfies all six aforementioned constraints, and how does this representation differ from existing methods? (**Q2**) How does the proposed material representation perform in material property prediction tasks? (**Q3**) How effective is our proposed multiple LLM agent architecture in generating graph representation code when facing new constraints?

### 5.1. A Representation Obtained by Codes of Rep-CodeGen

In this section, we answer question Q1. First, we introduce the results of Rep-codeGen. Then, we provide a detailed explanation of the material graph representation that satisfies six constraints. Finally, we thoroughly discuss the differences between our proposed representation and existing graph representation methods.

**Results of Rep-CodeGen.** The framework employs codes satisfying three constraints as bases and iterates until a material representation that passes all test cases (i.e., satisfies

all six constraints) is found. Ultimately, in the 14,076th iteration, we identified code that successfully passed all test cases. In the process of finding codes that satisfy six constraints, the framework generated a total of 56,304 codes, of which 39.60% (22,297) were executed successfully. Among these, 4,009 codes satisfied four constraints, 14,645 codes satisfied five constraints, and 1 code satisfied six constraints. These results demonstrate that Rep-CodeGen is capable of generating code representations that satisfy new constraints.

Note that we employ three pre-trained QWen (Hui et al., 2024) LLMs as distinct agents, and the entire process does not require further retraining of the models and the results on other LLMs are shown in section 5.3.

**Material Graph Representation.** The material representation code obtained by Rep-CodeGen is provided in Appendix B. Specifically, the proposed representation that satisfies the physical constraints can be expressed as $[u, v, \text{attr}]$, where $u$ represents the atomic number of all atoms in the unit cell, and $v$ represents the atomic number of each neighbor of atom $u$. The attribute $\text{attr} = \{d_{uv}, \theta_{uv}\}$ includes the distance and dihedral angle information between atoms $u$ and $v$. The illustrations of $d_{uv}$ and $\theta_{uv}$ are shown in Figure 3 (j) and (k), where $d_{uv}$ represents the edge distance between atom $u$ and atom $v$, with atom $v$ being any neighbor of atom $u$. $\theta_{uv}$ represents the dihedral angle between the half-planes $uvk$ and $uvl$. This angle is also related to the direction of normal vectors of the two half-planes and vector $uv$. Here, atoms $k$ and $l$ are obtained by extending atom $u$ along the lattice direction.

**Differences from Existing Graph Representation Methods.** The proposed representation differs from existing

*Table 1.* Comparison between our proposed material representation and other baselines in terms of test MAE on JARVIS dataset and The Materials Project dataset. The best results are shown in **bold**, and the second best results are shown with underlines. The number in parentheses after the algorithm represents the number of constraints that the algorithm satisfies.

| Method | JARVIS | | | | Materials Project | |
| --- | --- | --- | --- | --- | --- | --- |
| | Energy Hull | Total Energy | BandGap | Formation Energy | Formation Energy | BandGap |
| | meV | meV/atom | meV | meV/atom | meV/atom | meV |
| CGCNN (4) | 170 | 78 | 410 | 63 | 31 | 292 |
| ALiGNN (4) | 76 | 37 | 310 | 33.1 | 22.1 | 218 |
| MatFormer (4) | 64.2 | 35 | 300 | 32.5 | 21 | 211 |
| PotNet (4) | 55.4 | 32.4 | 273 | 29.4 | 18.8 | 204 |
| PerCNet (5) | 50.3 | 30.7 | 265 | 28.7 | 18.1 | 200 |
| ComFormer (5) | 47 | 28.8 | 260 | **27.2** | 18.3 | 193 |
| Ours (6) | **43.3** | **28.3** | **253** | **27.2** | **18.0** | **191** |

graph algorithms primarily in two aspects: the method of acquiring neighbors $v$ and the construction of features attr. Below, we illustrate the distinctions of our representation by comparing it with the most constrained graph representation methods, namely ComFormer and PerCNet.

First, regarding the method of acquiring neighbors, existing methods, such as ComFormer(Yan et al., 2024) and PerCNet(Huang et al., 2025) in Figure 3, predominantly determine neighbor relationships based on a cutoff distance. Specifically, atoms within a distance smaller than the cutoff are considered neighbors. This approach lacks Lipschitz continuity for atoms near the cutoff boundary, meaning that minor positional changes of boundary atoms can lead to significant alterations in neighbor relationships. In contrast, our representation innovatively identifies neighbors based on periodicity. For any atom within a unit cell, all atoms within a $3 \times 3 \times 3$ periodic region surrounding the unit cell are considered neighbors, ensuring continuity to positional variations.

Second, in terms of feature construction, ComFormer focuses on distance and angle information, while PerCNet incorporates distance, angle, and dihedral angle information. Our representation, on the other hand, emphasizes distance and dihedral angle information. Notably, although both our representation and PerCNet model dihedral angles, our construction method differs significantly. As shown in Figure 3, in PerCNet's representation, among the four atoms forming two faces, the two atoms besides atom $u$ and its neighbor $v$ are selected based on their distance to atom $u$. This distance-based selection still suffers from the lack of Lipschitz continuity. In contrast, our representation selects the remaining two atoms based on lattice coordinates related to periodicity. Consequently, even if atomic coordinates fluctuate, the derived distance and dihedral angle information maintain Lipschitz continuity.

Detailed proofs demonstrating that our representation satisfies all constraints proposed in Section 3.2, as well as the role of the aforementioned distinctions, are provided in Appendices E.1 through E.6. To the best of our knowledge, this material representation is the first to simultaneously satisfy all of these constraints.

## 5.2. Performance Comparisons on Material Property Prediction tasks

In this section, we aim to answer question Q2 by evaluating the performance of the proposed presentation in terms of Mean Absolute Error (MAE), consistent with prior studies (Huang et al., 2025; Lin et al., 2023; Yan et al., 2022; Xie & Grossman, 2018).

**Setup.** The Materials Project-2018.6 dataset contains 69,239 materials, and the JARVIS-DFT-2021.8.18 3D dataset contains 55,722 materials. To ensure a fair comparison, we adopt the same data settings as previous works (Huang et al., 2025; Xie & Grossman, 2018). The statistics of data settings and neural network configurations are shown in the Appendix F.1.

**Results.** The results of various graph-based representation algorithms on the property prediction task are shown in Table 1, and point cloud-based algorithms are shown in Table 5 in the Appendix. Across all tasks, based on the material representation proposed by our multiple-agent architecture, our method achieves the best performance. Notably, while we employ the same network architecture as PerCNet for processing graph representations, our method demonstrates improved prediction accuracy across all tasks compared to PerCNet. The most significant improvement is observed in the prediction accuracy of Energy Hull, with an increase of 13.9%. This result highlights that the performance gains stem from the superiority of our representation rather than the neural network architecture.

*Table 2.* Constraint satisfaction results of representation codes generated by different large models under different frameworks. Algorithm variants with (1e3) denote 1,000 candidate code generations. The goal of this experiment is to satisfy five constraints. **Bold** values highlight bond metric superiority of our Rep-CodeGen framework over direct generation baselines.

| Algorithm | Improve Rate | Constraint Satisfaction Spectrum | | | | | |
|---|---|---|---|---|---|---|---|
| | | Zero | One | Two | Three (Initial) | Four | Five |
| GPT 3.5 (1e3) | 0.00% | 0.00% | 0.00% | 14.39% | 85.61% | 0.00% | 0.00% |
| GPT + Rep-CodeGen (1e3) | **5.41**% | 2.70% | 10.81% | 5.41% | 75.68% | **5.41%** | 0.00% |
| Deep Seek (1e3) | 0.00% | 0.00% | 0.00% | 0.00% | 100.00% | 0.00% | 0.00% |
| Seek + Rep-CodeGen (1e3) | **26.27**% | 0.00% | 0.00% | 0.42% | 73.31% | **26.27%** | 0.00% |
| QWen (1e3) | 13.51% | 2.70% | 2.70% | 10.81% | 70.27% | 13.51% | 0.00% |
| Qwen + Rep-CodeGen (1e3) | **66.53**% | 1.15% | 1.54% | 6.92% | 23.85% | **40.38%** | **26.15%** |

## 5.3. Performance Comparison on Evolution Ability of Rep-CodeGen

In this section, we aim to address question Q3 by conducting a comparative analysis between two distinct experimental conditions: the performance of large language models (LLMs) when augmented with Rep-CodeGen versus their baseline capabilities in standalone operation.

**Setup.** We conduct experiments on three different large language models: QWen (Hui et al., 2024), DeepSeek (DeepSeek, 2025), and ChatGPT-3.5 (OpenAI, 2023). Each LLM is evaluated on graph representation code generation tasks under two operational settings: (1) LLM solely, (2) LLM enhanced with Rep-CodeGen. To ensure a fair comparison, identical prompt templates are applied across corresponding configurations, with the baseline LLM prompting strategy detailed in Appendix F.2. Notably, in contrast to the aforementioned experiments, our objective is to identify representations that satisfy five known constraints, with the number of evolutionary iterations restricted to 1000 due to the resource limitation. As the code of graph representations satisfies three constraints is used as bases, the framework shows evolutionary ability if the generated codes of graph representations satisfy four and five constraints. By contrast, the framework shows degeneration if the generated codes of graph representations satisfy two or fewer constraints.

In this section, we employ two complementary metrics: **Improve Rate**: The proportion of executable codes that satisfy more constraints than the initial code. **Constraint Satisfaction Spectrum**: Distribution of codes satisfying incremental constraint counts (0-5) among successful executions.

**Results.** As shown in Table 2, our framework significantly improves the quality of generated code compared to directly using LLMs. For GPT3.5, we can see the quality of codes shows a degeneration to satisfy the two constraints, while GPT3.5 with Rep-CodeGen shows both evolution and degeneration. Note that, both the degeneration and evolution show the exploration ability, which is key to the success of generation models. The Deepseek shows no exploration ability for this task and fails to identify satisfied codes. By contrast, the Deepseek with Rep-CodeGen shows the exploration ability and finds the codes satisfying four constraints. Finally, the QWen shows the exploration ability and evolution ability. However, it can't generate codes satisfying five constraints. By contrast, Rep-CodeGen can significantly improve the abilities of QWen and generate codes satisfying five constraints. Overall, Rep-CodeGen demonstrates evolutionary capabilities across all three LLMs, supported by its enhanced exploration ability.

## 6. Conclusion

In this work, we focus on the challenge of obtaining graph representations of materials that are effective when facing new constraints. We propose Rep-CodeGen with iterative stages, a code generation framework with multiple LLM agents, to generate high-quality codes to obtain graph representations. To the best of our knowledge, Rep-CodeGen is the first framework for automatically generating codes to obtain graph representations. We take representation construction task as a code-generation task, which can reduce expert knowledge to a large extent. The researchers in material science can focus more on investigating the constraints brought by materials, while the researchers in computer science can focus on improving the quality and efficiency of generating methods. Furthermore, a type of representation from generated codes by our framework satisfies six constraints with codes satisfying three constraints as bases. We detailed analysis the difference between the obtained graph representations and existing graph representations, and provide chemical insights into such representations. Extensive experiments on two real-world material datasets show that a property prediction method based on such graph representations achieves state-of-the-art performance in material property prediction tasks. Furthermore, to evaluate the generalizability of our framework, we conduct experiments on different LLMs. The Rep-CodeGen shows evolution ability based on different LLMs, while the single LLM agent shows limited ability when facing new constraints. In the future, the proposed framework will be utilized to streamline material structure generation tasks.

## Acknowledgements

This work was supported by the National Natural Science Foundation of China underGrant Nos.U22A2098, 62172185, 62206105, 62202200, and 62406127; the Major Science and Technology Development Plan of JilinProvince under Grant No. 20240302078GX; and the Major Science and Technology Development Plan of Changchun under Grant No.2024WX05.

## Impact Statement

This paper presents work whose goal is to advance the filed of AI4Science, especially for material science. There are many potential societal consequences of our work, none which we feel must be specifically highlighted here.

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

## A. The fulfillment of physical constraints of related works

The fulfillment of physical constraints by various algorithms is shown in Table 3.

Table 3. The fulfillment of physical constraints by various algorithms, where × indicates that the algorithm does not satisfy the constraint, and ✓ indicates that it does.

| Algorithm | Permutation Invariance | Rotation Invariance | Reflection Equivariance | Lipschitz Continuity | Periodicity Equivariance | Translation Invariance |
|---|---|---|---|---|---|---|
| CGCNN | ✓ | ✓ | × | × | ✓ | ✓ |
| ALiGNN | ✓ | ✓ | × | × | ✓ | ✓ |
| MatFormer | ✓ | ✓ | × | × | ✓ | ✓ |
| PotNet | ✓ | ✓ | × | × | ✓ | ✓ |
| materialLLM | × | × | × | × | ✓ | × |
| MatExpert | × | × | × | × | ✓ | × |
| PerCNet | ✓ | ✓ | ✓ | × | ✓ | ✓ |
| ComFormer | ✓ | ✓ | ✓ | × | ✓ | ✓ |
| AMD | ✓ | ✓ | × | ✓ | ✓ | ✓ |
| PDD | ✓ | ✓ | × | ✓ | ✓ | ✓ |
| Ours | ✓ | ✓ | ✓ | ✓ | ✓ | ✓ |

## B. Material representation code proposed by Rep-CodeGen

The code of material representation generated by our proposed Material Representation Code Generation (Rep-CodeGen) with multi-agent architecture is shown in Code 1. To enhance readability, we have manually adjusted the variable naming conventions in the generated code to align with those used in the paper and added necessary comments, while keeping other parts unchanged.

Listing 1. Code of material representation generated by code generation framework with multiple LLM agents (Rep-CodeGen)

```
@funsearch.evolve                                                          1
def representation(dataset_list: list) -> list:                            2
  atoms_Attr_list = []                                                     3
  for atoms_data in dataset_list:                                          4
    coords = np.array(atoms_data["coords"])                                5
    elements = atoms_data["elements"]                                      6
    lattice_mat = np.array(atoms_data["lattice_mat"])                      7
    num_atoms = len(coords)                                                8
    extended_atoms = []                                                    9
                                                                           10
    # Extend atoms based on lattice translations                          11
    for dx in [0, -1, 1]:                                                  12
      for dy in [0, -1, 1]:                                                13
        for dz in [0, -1, 1]:                                              14
          for i in range(num_atoms):                                       15
            coord_u = coords[i]                                            16
            element_u = elements[i]                                        17
            new_coord = np.array(coord_u + dx * lattice_mat[0] + dy * lattice_mat[1] + dz   18
                * lattice_mat[2])
            extended_atoms.append([elements[i], new_coord])                19
                                                                           20
    Attr_list = []                                                        21
    for i in range(num_atoms):                                             22
      coord_u = extended_atoms[i][1]                                       23
      u = element_to_atomic_number[extended_atoms[i][0]]                   24
      distances = []                                                       25
                                                                           26
      # Calculate distances to all other atoms                            27
      for j in range(len(extended_atoms)):                                 28
        if i != j:                                                         29
          coord_v = extended_atoms[j][1]                                   30
```

```
          v = element_to_atomic_number[extended_atoms[j][0]]              31
          distance = np.linalg.norm(coord_u - coord_v)                   32
          distances.append([j, distance])                                33
      distances.sort(key=lambda x: x[1])                                 34
      nearest_neighbors = distances[:]                                   35
                                                                         36
      # Process nearest neighbors                                        37
      for neighbor in nearest_neighbors:                                 38
          j = neighbor[0]                                                39
          distance = neighbor[1]                                         40
          u = element_to_atomic_number[extended_atoms[i][0]]             41
          v = element_to_atomic_number[extended_atoms[j][0]]             42
          Attr = np.round(distance, 2)                                   43
          Attr_list.append([u, v, Attr])                                 44
                                                                         45
          # Calculate dihedral angle                                     46
          u_coord = extended_atoms[i][1]                                 47
          v_coord = extended_atoms[j][1]                                 48
          k_coord = extended_atoms[i + num_atoms][1]                     49
          l_coord = extended_atoms[i + 3 * num_atoms][1]                 50
                                                                         51
          uv = np.array(v_coord) - np.array(u_coord)                     52
          uk = np.array(k_coord) - np.array(u_coord)                     53
          ul = np.array(l_coord) - np.array(u_coord)                     54
          normal_ukl = np.cross(uk, ul)                                  55
          normal_uvk = np.cross(uv, uk)                                  56
          cos_angle = np.dot(normal_ukl, normal_uvk) / (np.linalg.norm(normal_ukl) * np.  57
              linalg.norm(normal_uvk))
          cos_angle = np.clip(cos_angle, -1.0, 1.0)                      58
          angle = np.arccos(cos_angle)                                   59
          cross_prod = np.cross(normal_ukl, normal_uvk)                  60
          if np.dot(cross_prod, uv) > 0:                                 61
              dihedral_angle = np.degrees(angle)                         62
          else:                                                          63
              dihedral_angle = np.degrees(np.pi - angle)                 64
          Attr_list.append([u, v, dihedral_angle])                      65
                                                                         66
      Attr_list.sort(key=lambda x: (x[0], x[1], x[2]))                   67
      atoms_Attr_list.append(Attr_list)                                 68
                                                                         69
  return atoms_Attr_list                                                 70
```

## C. The construction of test sets for each constraint

- **Permutation Invariance**: For each material structure in the mp-20 dataset, we rearrange the order of all the atoms. If the representation of the material structure changes after rearranging, it is considered that the representation does not satisfy permutation invariance.

- **Rotation Invariance**: For each material structure in the mp-20 dataset, we rotate all the atomic coordinates and lattice coordinates within the unit cell by a random angle. If the representation of the material structure changes after rotation, it is considered that the representation does not satisfy rotation invariance.

- **Translation Invariance**: For each material structure in the mp-20 dataset, we translate all the atomic coordinates within the unit cell by the same random vector. If the representation of the material structure changes before and after translation, it is considered that the representation does not satisfy translation invariance.

- **Reflection Equivariance**: For each material structure in the mp-20 dataset, we perform a mirror reflection of all the atomic coordinates and lattice coordinates along a randomly chosen plane. If the representation before and after the reflection remains the same, it is considered that the representation does not satisfy reflection equivariance.

- **Periodicity Equivariance**: For each material structure in the mp-20 dataset, we modify the lattice coordinates in various ways such that the modified coordinates cannot be obtained by rotating or translating the original lattice

coordinates. If the representation of the material structure before and after the modification remains unchanged, it is considered that the representation does not satisfy periodicity equivariance.

- **Lipschitz Continuity**: For each material structure in the mp-20 dataset, we make small changes to the coordinates of some atoms in the unit cell. If the material representation undergoes abrupt changes (such as a change in neighbor relationships) or the difference between the representation before and after modification is too large, it is considered that the representation does not satisfy Lipschitz continuity.

## D. Prompt of all agents

### D.1. Prompt of $Agent\_A$

An example template of the prompt of $Agent\_A$ is :

```
I want to design a function that can construct a graph representation
of material molecules that satisfies specific constraints.  The graph
representation of the material molecule should meet the following requirements:
<Symmetry_Constraints>.  \n Below is the code that describes this problem, which
does not require modification:  <Problem_Code>.\n Next is the function that
needs modification:<Representation_Code>.\n The constraint satisfaction of the
material graph representations generated by the above representation function
is as follows:  <Satisfaction_Reason>.\n Your task is to outline a clear plan
and detailed steps to design a new representation function that fulfills all the
specified constraints.
```

### D.2. Prompt of $Agent\_B$

An example template of the prompt of $Agent\_B$ is :

```
I want to design a function that can construct a graph representation
of material molecules that satisfies specific constraints.  The graph
representation of the material molecule should meet the following requirements:
<Symmetry_Constraints>.  \n Below is the code that describes this problem, which
does not require modification:  <Problem_Code>.\n Next is the function that needs
optimization:  <Representation_Code >.  \n I hope to rewrite a new representation
function to replace the previous representation function, so that the graph
representation it generates satisfies all constraints.  Here is my proposed plan:
<Modification_Plan >.  \n Your task is to rewrite a new representation function
to replace the previous one, based on the plan outlined above.
```

### D.3. Prompt of $Agent\_C$

An example template of the prompt of $Agent\_C$ is :

```
I want to design a function that can construct a graph representation
of material molecules that satisfies specific constraints.  The graph
representation of the material molecule should meet the following requirements:
<Symmetry_Constraints>.  \n Below is the code that describes this problem, which
does not require modification:  <Problem_Code>.\n Next is the function that needs
optimization:  <Representation_Code >.  \n The constraint satisfaction of the
material graph representations generated by the above representation function is
as follows:  <constraint_satisfaction >.  \n Your task is to analyze the reasons
for each constraint being satisfied or not satisfied, one by one.  Please present
the complete reasoning.
```

### D.4. Description of Constraints

Below is the full description of the constraints:

- **Permutation invariance**: Changing the atomic indices should not alter the graph representation.

- **Rotation invariance**: Rotating the atomic coordinates should not change the graph representation.

- **Reflection equivariance**: Performing a mirror symmetry on the lattice and atomic coordinates should result in a change in the graph representation.

- **Lipschitz continuity**: If the coordinates of the crystal undergo continuous changes, the corresponding graph representation should also change continuously. This means the neighbor relationships of each atom (i.e., which atom is a neighbor) should not change, but the attributes of the corresponding neighbors (such as distance or angle) should vary accordingly.

- **Periodicity equivariance**: The graph representation should implicitly incorporate lattice information, meaning that any modification to the lattice coordinates (e.g., scaling the lattice by 1.5 times) should result in a change in the graph representation.

- **Translation invariance**: Translating the atomic coordinates should not affect the graph representation.

## E. Proof that the proposed representation satisfy material symmetry properties

Below, we will prove that the material representation found by Rep-CodeGen satisfies all the constraints proposed in Section 3.2, based on the representation formula in Section 5.1.

### E.1. Permutation Invariance

As shown in Section 5.1, the features of atom $u$ and its neighboring atom $v$ depend solely on the atomic indices. For example, if atom $u$ is a carbon (C) atom, its feature value would be 6. Therefore, changing the arrangement of atoms does not affect the final representation.

### E.2. Rotation Invariance

The variables in `attr`, which contain distance and dihedral angle information, are relative quantities. Therefore, applying rotation to the coordinates does not affect the material representation.

### E.3. Reflection Equivariance

When the coordinates of the material undergo reflection, the directions of the vectors within the material will change. As a result, the product of the variables `cross_prad` and `uv` on line 61 of Code 1 will be altered. Therefore, the representation we obtain satisfies reflection equivariance.

### E.4. Lipschitz Continuity

The feature triplet $[u, v, \texttt{attr}]$ for each atom within the unit cell exhibits continuity primarily in two aspects: the feature value `attr` and the neighbor relationship $v$. Since `attr` is directly related to the coordinates, it inherently satisfies continuity. The current method, however, employs a cutoff approach, which can lead to abrupt changes in the neighbor relationships. In contrast, the representation we propose considers all atoms within a $3 \times 3 \times 3$ lattice as neighbors, ensuring that the neighbor relationships do not change due to small variations in the coordinates, thereby satisfying Lipschitz continuity.

### E.5. Periodicity Equivariance

For any atom $u$ within the unit cell and its neighboring atom $v$, the coordinates of $v$ are dependent on both the atomic coordinates within the lattice and the lattice coordinates. Therefore, when the periodicity of the material changes, i.e., when the lattice coordinates are modified, the coordinates of neighbor $v$ will also change. As a result, the distance feature between $u$ and $v$ will be adjusted accordingly, which ensures that the material representation satisfies periodicity equivariance.

## E.6. Translation Invariance

The variables in `attr`, which contain distance and dihedral angle information, are relative quantities. Therefore, applying translation to the coordinates does not affect the material representation.

*Table 4.* Statistics of datasets.

| Dataset | JARVIS | | | | Materials Project | |
|---|---|---|---|---|---|---|
| | Formation Energy | Total energy | Bandgap | Ehull | Formation Energy | BandGap |
| # training | 44578 | 44578 | 14537 | 44296 | 60000 | 60000 |
| # validation | 5572 | 5572 | 1817 | 5537 | 5000 | 5000 |
| # testing | 5572 | 5572 | 1817 | 5537 | 4239 | 4239 |

# F. Experimental Details

## F.1. Experimental Setup of material Property Prediction tasks

The statistics of data settings are shown in Table 4. For all tasks, we utilize one NVIDIA RTX 24G 3090 GPU for computation. To ensure a fair comparison, we employ the same network architecture and training methodology as PerCNet(Huang et al., 2025), which also incorporates dihedral angle information. In terms of implementation, all models are trained using the Adam optimizer with a one-cycle learning rate scheduler. The training configuration includes a learning rate of 0.001, a batch size of 64, and a total of 500 training epochs.

## F.2. Prompt of LLM without architecture assistance

An example template of the prompt of LLM is :

```
I want to design a function that can construct a graph representation
of material molecules that satisfies specific constraints.  The graph
representation of the material molecule should meet the following requirements:
<Symmetry_Constraints>.  \n Below is the code that describes this problem, which
does not require modification:  <Problem_Code>.\n Next is the function that needs
modification:<Representation_Code>.\n Your task is to provide a new, modified
function that will replace the current priority function.  This new function
should generate a graph representation that satisfies all the given constraints,
and the function signature should remain the same as the original priority
function.  Please ensure that the new function adheres to all the necessary
constraints and provides a correct implementation.  Only the full code of the
new function should be returned, with no additional comments, explanations, or
suggestions.
```

## F.3. Comparison with point cloud-based representation methods

The results of point cloud-based algorithms are shown in Table 5.

*Table 5.* Comparison between our proposed material representation and other baselines in terms of test MAE on JARVIS dataset in formation energy task. The results of baselines are from related work (Keqiang et al., 2024)

| Method | Number of training materials | Test MAE (eV/atom) |
|---|---|---|
| **PDD**(Widdowson & Kurlin, 2022) | 29342 | 0.047 |
| **AMD**(Widdowson et al., 2022) | 24067 | 0.78 |
| **Ours** | 24067 | 0.027 |

