# OpenReview forum: "Code-Generated Graph Representations Using Multiple LLM Agents for Material Properties Prediction"
_ICML.cc/2025/Conference — ICML 2025 poster_

### Official Review · Reviewer_oQXj · 2025-02-16

**Overall Recommendation:** 3

**Summary:**

This paper presents Rep-CodeGen, a framework using multiple LLM agents to generate, evolve, and evaluate code for obtaining graph representation of crystal structures following physical constraints. The representation obtained thereby is tested for constraint satisfaction and performance in materials property prediction, in comparison to baseline methods.

## update after rebuttal
The authors have answered my questions well. I've raised my score accordingly; however, my concern remains that proposing a new representation is not a task for which LLM could be very valuable. The contribution to materials science is limited.

**Claims And Evidence:**

The main claims, which are related to the 3 key questions in Sec. 5, are well-supported. However, my concerns are about the validity of 6 constraints and the usefulness of this Rep-CodeGen method (see Other S/W and Questions).

**Essential References Not Discussed:**

Related works to my knowledge are comprehensively discussed.

**Experimental Designs Or Analyses:**

Experimental design and analyses are sound. The comparison of evolution ability (Sec. 5.3) needs a little clarification (see Questions).

**Methods And Evaluation Criteria:**

The methods for comparing materials property prediction performance and testing constraint satisfactions make sense.

**Other Comments Or Suggestions:**

None.

**Other Strengths And Weaknesses:**

This work uses LLM to “create” a new graph representation through code generation. However, designing this new representation and is not beyond human’s capability given the goal of which constraints to satisfy. Once a function for generating such representation is written, it can be reused for all materials. There are not many constraints of this kind, so this “representation creation” process will not need to be repeated so many times that it needs an LLM to automate (not to mention using LLMs decreases trustworthiness). Given these, I don’t see an advantage of using LLM here.

**Questions For Authors:**

* Materials representations need to trade off conciseness and expressiveness. Sec. 3.2 lists 6 types of constraints, but are they all necessary for representing materials?
  - Enforcing reflection equivariance (rather than invariance) helps distinguish chirality, which has little or no impact on the properties tested in this study but increases the complexity of representation.
  - For crystals with lattice type and atomic radii known, a cutoff radius for formulating graphs could be reasonably chosen. Whereas enforcing Lipschitz continuity may result in different structures with a small distortion (e.g., Jahn Teller) not well distinguished.
* Lots of different code pieces are generated during the evolution process, some may contain certain merits that others miss. Is there a systematic way to combine them?
* In Sec. 5.3, is the “base” code provided by user or generated by each LLM from prompt?
* (Just for curiosity) Why choose to generate code instead of natural language description?

**Relation To Broader Scientific Literature:**

Creating new representations of materials structures that satisfy *necessary* physical constraints could help ML-based materials modeling and design.

**Theoretical Claims:**

The proofs of constraint satisfactions look correct. However, whether all constraints in Sec. 3.2 are necessary is questionable (see Questions).

---

> ### Author Rebuttal · Authors · 2025-03-27
>
> We sincerely appreciate your time and thoughtful evaluation, as well as your recognition of our methodological design, theoretical foundations, and experimental framework. We are particularly grateful for your recognition of our motivation, ”Creating new representations of materials structures that satisfy necessary physical constraints could help ML-based materials modeling and design”. Below, we will address your concerns point by point.
>
> Q1: Materials representations need to trade off conciseness and expressiveness. Are six types of constraints all necessary?
>
> R1: We agree that representations should balance simplicity and expressiveness. At the same time, the importance of the six types of constraints proposed in this paper has been substantiated in corresponding studies (ComFormer, PerCNet, and PDD e.t.). These constraints can help capture the structure information of crystals and enhance the accuracy of property prediction. Improving the prediction accuracy of ML-based methods can benefit material science in various aspects.
>
> Most importantly, our framework offers fully customizable constraints for crystal representation. Domain experts can freely modify these constraints, such as physical constraints, representational simplicity constraints, and other specific constraints, based on application needs, enabling the creation of tailored crystal characterization codes for various scenarios. We selected six representative constraints to demonstrate our framework's capability in generating crystal representations that simultaneously satisfy multiple constraints that existing algorithms fail to meet collectively.
>
> Q2: How to combine merits from different code pieces? Is the ‘base’ code provided by user?
>
> R2: In the initial iteration, we manually provide the 'base' code to Agent_A. This code is designed to meet fundamental properties, including permutation, rotation, and translation invariance. It is worth noting that the base code can alternatively be generated by LLMs.
>
> The merits of the generated codes are integrated throughout our framework's process. We not only retain the codes (produced by Agent_B) that meet specific constraints but also document the reasoning (merits and demerits) by Agent_C. Then, the parent codes with merits can be effectively selected through Formula 1 and 2. Subsequently, the codes and their corresponding reasoning are sent back to Agent_A. Agent_A evaluates the codes and their rationale, devising an improved plan accordingly.
>
> Q3: Why choose to generate code instead of natural language description? (Using LLMs decreases trustworthiness.)
>
> R3: This question helps shed light on our motivation. The primary advantage of code lies in its superior clarity and interpretability. Its structured nature allows humans to easily understand the reasoning behind the LLM's outputs, thereby boosting trust in its results. In contrast, natural language descriptions of crystal structures lack the same level of transparency, making it harder to discern the rationale behind the LLM's conclusions and potentially undermining confidence.
>
> Q4: “Given these, I don't see an advantage of using LLM here.”
>
> R4: It is worth noting that our goal is to assist human experts in accelerating material research instead of completely replacing them. This acceleration manifests in two aspects.
>
> First, while human experts excel in their specialized fields, they often invest significant effort in acquiring cross-disciplinary skills, such as coding or understanding graph learning. Rep-CodeGen enables materials science professionals to concentrate on their core expertise by reducing the need to master complex computer science concepts, thereby saving both time and resources.
>
> Second, the generated representations can be viewed as a novel source of knowledge that can inspire the researchers. Rep-CodeGen shows the capability to rapidly generate a diverse array of solutions. For instance, to ensure that representations satisfy periodic equivariance (i.e., changes in lattice coordinates should be reflected in the representation), current graph representation methods typically rely on obtaining sufficiently large cutoffs. By contrast, RepCode-Gen introduces innovative strategies beyond the traditional cutoff method, such as converting periodic data into length and angle metrics, incorporating angles between lattice and edge vectors, and proposing supercell configurations extending unit cells along periodic dimensions.

---

> > ### Comment · Reviewer_oQXj · 2025-04-03
> >
> > I appreciate the Authors' time and effort. My concerns regarding clarity are well addressed; I see a good point of generating code instead of natural language. My concern remains that proposing a new representation is not a repetitive/laborious task which LLM could be very valuable to. I'll change my rating accordingly.

---

### Official Review · Reviewer_DgpU · 2025-02-26

**Overall Recommendation:** 4

**Summary:**

This paper introduces a novel framework named Rep-CodeGen, which leverages multiple Large Language Model (LLM) agents to automatically generate code for obtaining graph representations of material properties. The primary contributions of this work are threefold. First, the paper proposes an interpretable framework capable of automatically generating code to derive graph representations, particularly effective when addressing new constraints. This represents the first framework of its kind to automate the generation of code for graph representations. Second, through the generated code, the paper achieves a graph representation that satisfies six distinct constraints, including permutation invariance, rotation invariance, reflection equivariance, Lipschitz continuity, periodicity equivariance, and translation invariance. Lastly, extensive experiments on two real-world material datasets demonstrate that the material property prediction method based on this graph representation achieves state-of-the-art performance across multiple tasks.

## update after rebuttal
After reviewing the authors’ rebuttal, my questions regarding experimental details, the evolution process, and constraint handling have been satisfactorily addressed. I have no further concerns and maintain my recommendation of Accept.

**Claims And Evidence:**

Yes. Specifically, the claims made in the paper can be divided into four parts. First, the paper asserts that Rep-CodeGen is the first framework for automatically generating codes to obtain representations that can be used when facing new constraints. This claim is supported by the fact that no prior work has adopted this approach for generating crystal representations, as demonstrated in Section 2.1. Second, the paper claims that the framework is indeed capable of generating representations that satisfy new constraints. This claim is substantiated by the experimental results presented in Sections 5.1 and 5.3. Third, the paper proposes a representation that satisfies six constraints using the Rep-CodeGen framework. This claim is validated in Section 5.1 and further supported by the code provided in Appendix Code 1 and the proof section. Finally, the paper claims that the obtained representations achieve state-of-the-art (SOTA) performance in crystal property prediction tasks, which is evidenced by the results of Experiment 2.

**Essential References Not Discussed:**

This paper have discussed all the essential references.

**Experimental Designs Or Analyses:**

Yes, the experimental designs and analyses presented by the authors are sound and valid. Specifically:

1. Experiment 1: The authors introduce the generation results of the Rep-CodeGen framework, compare the generated representations with other representations, and provide theoretical proofs demonstrating that the generated representations satisfy the proposed constraints.
2. Experiment 2: The authors evaluate the performance of the generated representations in crystal property prediction tasks, demonstrating their effectiveness.
3. Experiment 3: The authors compare the generation results of three different large language models (LLMs) when used independently versus in combination with the proposed framework, highlighting the advantages of their approach.
Overall, the experiments are well-designed, and the analyses are thorough and logically consistent.

**Methods And Evaluation Criteria:**

Yes. The authors' approach of using a multi-agent framework to generate crystal representation codes is well-founded and logical. Additionally, the experimental design is appropriate and aligns with the research objectives. The datasets and evaluation metrics employed for the crystal property prediction task are publicly available and widely recognized in the field, further validating the suitability of the proposed methods and criteria.

**Other Comments Or Suggestions:**

Suggestions:
1. It is recommended to provide the initial code to help readers better understand the extent of changes during the evolution process.
2. It is recommended to provide the complete prompts, particularly the descriptions of the constraints, to enhance reproducibility and clarity.

**Other Strengths And Weaknesses:**

Strengths:
1. Originality: The approach to solving the problem is novel. The paper utilizes the multi-agent framework Rep-CodeGen to generate codes for crystal representations, which, compared to previous manual design methods, has the capability to satisfy unknown new constraints. Moreover, the paper is the first to propose a representation that satisfies six constraints using the Rep-CodeGen framework.
2. Experimental Design: The experimental setup is reasonable, and the results demonstrate that the proposed representation achieves state-of-the-art (SOTA) performance in crystal property prediction tasks.

Weaknesses:
Some experimental details should be more clearly specified. See suggestions and questions.

**Questions For Authors:**

Questions:
1. Could there be situations where constraints conflict with each other, meaning that satisfying some constraints might make it impossible to satisfy others? If such cases arise, how does the proposed framework handle them? A clear explanation of this scenario and the resolution strategy would significantly impact the evaluation of the framework's robustness.

2. In the conclusion, the authors mention that the proposed representation method could also be applied to crystal generation. Could the authors provide more details on how this representation would be utilized in crystal generation tasks?

**Relation To Broader Scientific Literature:**

The key contributions of this paper are closely tied to the broader scientific literature on crystal property prediction.
Contributions to Literature:
1. This paper introduces a novel solution by proposing a multi-agent framework for automatically generating crystal representations that inherently satisfy the desired constraints. This approach represents a departure from traditional manual design methods.
 2. The proposed framework generates representations that differ from existing ones and, for the first time, satisfies six specific constraints, addressing a gap in prior research.
3. The experimental results demonstrate that the generated representations outperform existing methods in crystal property prediction tasks, achieving higher accuracy.

**Theoretical Claims:**

Yes. The authors provide proofs in the appendix demonstrating that the representations generated by the Rep-CodeGen framework satisfy the six proposed constraints.  No significant issues were identified in the reasoning or validity of the proofs.

---

> ### Author Rebuttal · Authors · 2025-03-27
>
> Thank you very much for your time and for recognizing the originality of our work, the contributions to the problem, and the experimental design. Below, we will address your suggestions and questions point by point.
>
> Q1: It is recommended to provide the initial code to help readers better understand the extent of changes during the evolution process.
>
> R1: Due to space limitations, we are unable to present the complete code here.The initial code will be provided in the appendix. In terms of design philosophy, the initial code determines neighbors of atoms within the unit cell based on a distance cutoff, and the features are limited to the distances between neighbors. However, the representation code that satisfies all conditions after the framework evolution differs in both the neighbor determination method and the features.
>
> Q2: It is recommended to provide the complete prompts, particularly the descriptions of the constraints, to enhance reproducibility and clarity.
>
> R2: We will include the complete prompts in the appendix. Below is our full description of the constraints:
>
> Permutation_invariance: Changing the atomic indices should not alter the graph representation.
>
> Rotation_invariance: Rotating the atomic coordinates should not change the graph representation.
>
> Reflection_equivariance: Performing a mirror symmetry on the lattice and atomic coordinates should result in a change in the graph representation.
>
> Lipschitz_continuity: If the coordinates of the crystal undergo continuous changes, the corresponding graph representation should also change continuously. This means the neighbor relationships of each atom (i.e., which atom is a neighbor) should not change, but the attributes of the corresponding neighbors (such as distance or angle) should vary accordingly.
>
> Periodicity_equivariance: The graph representation should implicitly incorporate lattice information, meaning that any modification to the lattice coordinates (e.g., scaling the lattice by 1.5 times) should result in a change in the graph representation.
>
> Translation_invariance: Translating the atomic coordinates should not affect the graph representation.
>
> Q3: Could there be situations where constraints conflict with each other, meaning that satisfying some constraints might make it impossible to satisfy others?
>
> R3: Theoretically, this situation is highly unlikely to occur because these physical constraints are grounded in real-world conditions, making conflicts almost improbable. Moreover, even if conflicts were to arise, the framework would evolve in a direction that satisfies the majority of the constraints.
>
> Q4: In the conclusion, the authors mention that the proposed representation method could also be applied to crystal generation. Could the authors provide more details on how this representation would be utilized in crystal generation tasks?
>
> R4: First, taking state-of-the-art algorithms in crystal generation tasks, such as CDVAE[1], DiffCSP[2] and DiffCSP++[3], as examples, the graph representation we propose can serve as input to denoising models and be directly applied to crystal generation tasks.
>
> Second, the “constraints” that Rep-CodeGen can solve, are not limited to physical constraints but can also include arbitrary requirements from human experts. Therefore, our framework can generate new crystal representation codes that meet the needs of various scenarios, such as sequence-based crystal representations required by LLMs.
>
> [1]T. Xie, X. Fu, O.-E. Ganea, R. Barzilay, T. Jaakkola, Crystal diffusion variational autoencoder for periodic material generation, arXiv preprint arXiv:2110.06197 (2021).
>
> [2]R. Jiao, W. Huang, P. Lin, J. Han, P. Chen, Y. Lu, Y. Liu, Crystal structure prediction by joint equivariant diffusion, Advances in Neural Information Processing Systems 36 (2024).
>
> [3]Jiao, Rui, et al. "Space Group Constrained Crystal Generation." The Twelfth International Conference on Learning Representations.

---

> > ### Comment · Reviewer_DgpU · 2025-04-04
> >
> > The author has answered my questions well. I don't have any more questions. I will keep my score as Accept.

---

### Official Review · Reviewer_RrQ6 · 2025-03-07

**Overall Recommendation:** 4

**Summary:**

This paper proposes a novel code generation framework for material property prediction. The LLM agents are employed to replace the human experts and automatically generate codes to process the CIF files fitting GNN-based models. After the processing, the obtained input vectors are called representations. Due to the symmetry of crystals, the representation should satisfy a few constraints. This paper selects six widely used constraints as the target of the generated representations. Note that the code used from the beginning satisfies three types of constraints. As the framework evolves, the generated representations can satisfy six types of constraint, which did not exist before. This ability is summarized as "obtain representations that can be used when facing new constraints" by authors. The experiments show how the generated codes process the CIF files. The main difference is the neighborhood selection compared to the expert-designed methods. Furthermore, the generated representations with an existing GNN-based property model can achieve the SOTA performance in property prediction tasks.

**Claims And Evidence:**

Yes. There are two important claims in this paper. First, the authors claimed they found a new graph representation for crystalline materials. Second, the authors claimed the proposed code-generation framework can satisfy new constraints.
The two claims are supported well by experimental results. For the first claim, the new representation is shown in Fig.3 compared with two methods. For the second claim, the results in Table 2 show the evolution results among different LLMs.

**Essential References Not Discussed:**

No important references are found to be missing.

**Experimental Designs Or Analyses:**

Yes. The experiments are designed in accordance with the work on predicting crystal properties.

**Methods And Evaluation Criteria:**

Yes. The proposed framework is tested on two widely used material datasets, named JARVIS and Material Project separately. And the LLM agents are tested on GPT3.5, DeepSeek, and QianWen.

**Other Comments Or Suggestions:**

typos: line 014 "crystalline material data..."

**Other Strengths And Weaknesses:**

The main strengths of this paper are as follows:
- Most existing studies based on LLMs focus on directly using these models to predict crystal structures, i.e., generating new structures. In contrast, this work explores how to generate representations of crystal structures using LLMs, which is both novel and fundamental.
- The framework is effective in generating new codes for the representation. And the results are interesting and impressive. In particular, the results shown in Figure 3 are especially noteworthy, as they may inspire human experts with their unique representation.

The weaknesses of this paper are as follows:
- Section 5.3 is not very convincing. The main focus of the proposed framework is to generate codes (representations) satisfying six constraints, but the experiments are stopped at five due to resource limitations.
- Although the generated codes are interpretable, the paper does not provide detailed insights into understanding the logic and structure of these codes. There is also no discussion on how to further optimize these codes to improve performance.

**Questions For Authors:**

1. Why the results of GPT+Rep-CodeGen and Seek+Rep-CodeGen are 0 for five constraints in Table 2? Does this mean that the framework is only effective for QianWen?

2. The experiments used specific network architectures of PerCNet. Can the impact of network choices on the results be further discussed?

3. The test set is constructed manually, will it be better to employ the LLMs to generate the test set?

4. Will the framework get stuck in local optima when facing more complex constraints? If so, how can mechanisms be designed to avoid this?

**Relation To Broader Scientific Literature:**

1. The idea of this work holds significant importance for both material and computer science. The test sets bridges the two fields. Normally, researchers need to investigate how to satisfy the constraints by sophisticated design. By contrast, this work converts the constraints to test sets and discards the representations that failed in test sets. In other words, this framework can not only solve the problem of material science but can also deal with problems in protein with desired test sets accordingly.

2. A novel graph-based representation that satisfies six constraints is found by the proposed framework, thereby achieving enhanced accuracy in crystal property prediction.

**Theoretical Claims:**

Yes, I have checked the proofs in E. The proofs show why the generated representation can satisfy the six constraints. Most methods based on graph representations(e.g. CGCNN, ALiGNN, ComFormer) do not satisfy the reflection equivariance and Lipschitz Continuity. How to define "too large" in Lipschitz Continuity?

---

> ### Author Rebuttal · Authors · 2025-03-27
>
> Thank you very much to the reviewers for your time and for recognizing the innovation of our method and its contributions to the field. We will revise the paper carefully. Below, we will summarize your reviews and provide a response to each point separately.
>
> Q1: The experiments are stopped at five due to resource limitations. Why the results of GPT+Rep-CodeGen and Seek+Rep-CodeGen are 0 for five constraints in Table 2? Does this mean that the framework is only effective for QianWen?
>
> R1: Although the choice of five constraints is employed due to the resource limits, it can show a clearer comparison between LLMs with and without our framework. The zero results for GPT+Rep-CodeGen and Seek+Rep-CodeGen occur because we capped the experiment at 1,000 programs to ensure a fair comparison. We should also pay attention to the results other than three constraints which also reflect the evolutionary capability of our framework. Thus, the framework is not limited to QWen and can also be applied to other LLMs.
>
> Q2: Although the generated codes are interpretable, the paper does not provide detailed insights into understanding the logic and structure of these codes. There is also no discussion on how to further optimize these codes to improve performance.
>
> R2: In this paper, the framework goal is to identify crystal representations that satisfy all constraints, and thus we treat the entire evolutionary process as a black-box procedure. Our framework enables better utilization of the interdisciplinary knowledge of LLMs while guiding them to optimize and generate code effectively. Since the entire process is automated by the LLM, not interpreting the intermediate codes does not affect the final outcome.
>
> Q3: The experiments used specific network architectures of PerCNet. Can the impact of network choices on the results be further discussed?
>
> R3: We use PerCNet because it is currently the only network architecture in this field capable of handling dihedral angles. We employ this network to ensure a fair comparison between the representations obtained by our framework and those of PerCNet. In the future, we plan to use LLMs to simultaneously generate both the representations and the corresponding network architectures, rather than solely relying on network architectures from other algorithms.
>
> Q4: The test set is constructed manually, will it be better to employ the LLMs to generate the test set?
>
> R4: Using LLMs to generate the test set is possible, but in terms of effectiveness, a manually designed test set would be better. Due to the hallucination issue of LLMs, they might generate answers that appear correct but are actually wrong. Since the test set determines the direction of evolution, a manually designed test set would be more reliable and effective.
>
> Q5: Will the framework get stuck in local optima when facing more complex constraints? If so, how can mechanisms be designed to avoid this?
>
> R5: It is unlikely to fall into local optima because our program selection method (i.e., Equation 1 in the paper) ensures that the two parent programs have a higher probability of satisfying different constraints, thereby maintaining the diversity of the parent programs.

---

### Official Review · Reviewer_qN76 · 2025-03-18

**Overall Recommendation:** 4

**Summary:**

This paper introduces Rep-CodeGen, a framework that uses multiple LLM agents to autonomously generate graph representations for material property prediction. Unlike traditional methods, Rep-CodeGen iteratively refines representations through crossover generation, evaluation summary, and parent selection, ensuring adaptability. The framework optimizes graphs to satisfy six material constraints, improving prediction accuracy. Experimental results demonstrate its superiority over conventional approaches. By integrating LLMs for automated representation learning, this work reduces reliance on domain expertise, offering a promising solution for AI-driven materials science.

## update after rebuttal
During the rebuttal period, I interacted with the authors. The authors have answered my questions well, so I decided to raise my score for the paper.

**Claims And Evidence:**

Yes.

**Essential References Not Discussed:**

No.

**Experimental Designs Or Analyses:**

Yes.

**Methods And Evaluation Criteria:**

Yes.

**Other Comments Or Suggestions:**

I keep up with the literature in this area.

**Other Strengths And Weaknesses:**

**Strength:**
The generated codes are more interpretable, allowing humans to read and edit them easily. This AI knowledge may further inspire human researchers.
This paper shows that a good representation can improve the prediction results with the same model.

**Weakness:**
Please see the questions.

**Questions For Authors:**

1. The neighborhood selection is the main difference between the 'new' representation and the existing methods. Why does this change affect the property predictions in Table 1？

2. The LLMs may trained on methods and their codes in Table 3. Thus, the 'new' representation may be a combination of the existing codes. Then, how can the framework satisfy the unseen constraints?

3. This framework costs a lot of tokens to generate a target code. How to compare the cost from both time and money to DFT calculations? Besides, what is the size of LLMs in Table 2? As the LLMs become more and more powerful, a new version of GPT or Deepseek may accomplish this task well.

**Relation To Broader Scientific Literature:**

The 'new' representations can benefit the material virtual screening.

**Theoretical Claims:**

Yes.

---

> ### Author Rebuttal · Authors · 2025-03-27
>
> Thank you for your valuable time and your recognition of our work's motivation, novel contributions, and experimental design. We provide point-to-point responses to your questions as follows.
>
> Q1: The neighborhood selection is the main difference between the 'new' representation and the existing methods. Why does this change affect the property predictions in Table 1?
>
> R1: The neighbor selection method in the generated representation enhances the graph's capability to capture long-range atomic interactions. As a result, the energy-related properties are improved.
> Specifically, the interactions between atoms are not limited to adjacent atoms, and long-range interactions can extend to distant atoms, which play a key role in the stability of crystals. Taking ionic crystals as an example, the Coulomb force between ions is a type of long-range interaction with a relatively large range of action. Cations and anions are attracted to each other through the long-range Coulomb force, which maintains the structural stability of the crystal. In our representation, neighbors are determined based on periodic regions instead of a cutoff value, which can cover the atoms over a longer distance. Besides, the long-range interactions also affect the electrical and thermal properties of crystal materials.
>
> Q2: The LLMs may train on methods and their codes in Table 3. Thus, the 'new' representation may be a combination of the existing codes. Then, how can the framework satisfy the unseen constraints?
>
> R2: LLMs can be trained not only on materials science research papers but also on works from other fields, such as protein design and drug discovery. With all these training resources, LLMs have a better understanding of atom interactions inside the materials. Therefore, we aim to guide the LLMs in generating the desired representations. To the best of our knowledge, the neighborhood selection for the new representation, which is based on periodic regions, has not been observed. As shown in Table 2, the representations satisfying four and five can be viewed as unseen constraints. Among these representations, we also observed 'new' representations that are different from existing methods.
>
> Q3. This framework costs a lot of tokens to generate a target code. How to compare the cost from both time and money to DFT calculations? Besides, what is the size of LLMs in Table 2? As the LLMs become more and more powerful, a new version of GPT or Deepseek may accomplish this task well.
>
> R3: The cost of Rep-CodeGen is minimal compared to DFT calculations. We utilize the Qwen2.5-Coder-7B model to develop three agents, opting for the 7B version over the Qwen2.5-Coder-32B primarily due to cost considerations. While the Qwen2.5-Coder models are open source, the operational expense of the 32B model is significantly higher than that of the 7B model. Additionally, the 7B model offers considerably faster inference times. Our aim is to create a framework accessible to a broad spectrum of researchers. Specifically, we only utilize one NVIDIA RTX 24G 3090 GPU for all experiments.
>
> A larger LLM can benefit the framework, but a single one may not accomplish the task well, as shown in Table 3. GPT-3.5 (gpt-3.5-turbo used in our experiments) and DeepSeek (DeepSeek-Coder-v2 used in our experiments) represent a larger general-purpose LLM and a larger code-generation LLM, respectively. Nonetheless, their performance remains unsatisfactory.

---

### Decision · Program_Chairs · 2025-05-01

**Decision:**

Accept (poster)

**Comment:**

All reviewers are positive about this paper. I agree with the reviewers that this paper has proposed a novel code generation framework with multiple large language model agents. This research direction is important and still at an early stage. Overall, this paper presents a solid work and is potentially useful to the community. I am confident that this paper can be accepted.